# Pathogenesis of Vascular Retinal Manifestations in *COVID-19* Patients: A Review

**DOI:** 10.3390/biomedicines10112710

**Published:** 2022-10-26

**Authors:** Elisa D’Alessandro, Aki Kawasaki, Chiara M. Eandi

**Affiliations:** 1Fondation Asile des Aveugles, Hôpital Ophtalmique Jules Gonin, University of Lausanne, 1004 Lausanne, Switzerland; 2Department of Surgical Sciences, University of Torino, 10126 Torino, Italy

**Keywords:** *COVID-19*, retina, vasculature, pathophysiology, mechanism

## Abstract

Pandemic infection secondary to coronavirus disease 2019 (*COVID-19*) had an important impact on the general population affecting not only respiratory tract but also many other organs. Ocular manifestations are quite common at the level of the anterior segment (conjunctivitis, dry eye), while posterior segment and, in particular, retinal findings are less frequent. In the retina, *COVID-19* is associated with vascular events. Since retinal arteries and veins represent an accessible window to the microvasculature of the rest of the body, a better understanding of the profile of retinal vascular occlusive events may help elucidate mechanisms of thrombo-occlusive complications in other organs in patients affected by *COVID-19*. In this review, we conducted a systematic literature search focused on retinal arterial and/or retinal venous manifestations. Twenty-one studies were included, describing a wide range of manifestations from mild signs like cotton wool spots, focal and flame-shaped hemorrhages, and vein dilation to more severe retinal artery and vein occlusions. Two principal pathogenetic mechanisms are considered responsible for these complications: a hypercoagulative state and a massive inflammatory response leading to a disseminated intravascular coagulation-like syndrome.

## 1. Introduction

Since the initial outbreak in Wuhan, China in December 2019, coronavirus disease 2019 (*COVID-19*) caused by severe acute respiratory syndrome coronavirus-2 (*SARS-CoV-2*) has rapidly spread around the world causing a global pandemic and a health and social crisis. The virus has a high tropism for the respiratory tract and can lead to respiratory distress syndrome, which can be fatal, especially in risk patients with comorbidities. However, it is now recognized that *COVID-19* affects many other systems and, using real-time reverse polymerase chain reaction (RT-PCR), *SARS-CoV-2* has been identified in several tissues throughout the human body [1]. *SARS-CoV-2* has also been detected in several types of clinical specimens, including tears and conjunctival swabs. Various ophthalmic manifestations of *COVID-19* have been reported with ocular surface disorders such as conjunctivitis, dry eye and epiphora being the most common [2,3]. Therefore, the eye has been considered as a potential route of entry for the virus.

*SARS-CoV-2* infects human cells by binding to the angiotensin converting enzyme 2 (ACE2) receptor and the transmembrane protease serine 2 (TMPRSS2), one of a group of membrane serine proteases that are expressed throughout the human body, especially by vascular endothelial cells [1]. At the ocular surface, the ACE2 receptor and the TMPRSS2 protease are expressed in the human cornea and conjunctival epithelium, especially the limbus, and common ocular manifestations are dry eye, epiphora and conjunctivitis.

Reports of posterior segment findings in *COVID-19* positive patients are less common. Recent studies have reported the presence of retinal microvascular abnormalities such as cotton wool spots, intraretinal hemorrhages, and vessel dilation in visually asymptomatic patients [4,5,6,7,8,9], while fewer reports describe retinal arterial or vein occlusion with visual impairment in *COVID-19* patients. The presence of cardiovascular risk factors in such patients are confounding factors for establishing a clear etiologic relationship between the retinovascular occlusive event and the *SARS-CoV-2* infection.

Through examination of the ocular fundus and fundus photography (which is fast and non-invasive), ophthalmologists have direct access to the retina and its vasculature. Retinal arteries and veins are considered to represent the state of the rest of the body’s microvascular system. Due to its accessibility, fundus examination is commonly used in screening for systemic vascular diseases, such as diabetes and hypertension [10]. *SARS-CoV-2* has been identified in the retina and it is known to determine a direct vascular endothelial damage, making the retinal examination a unique opportunity for studying the clinical consequences of *COVID-19* in vivo. Post-mortem eye examination of *COVID-19* patients are, however, fundamental in understanding the pathophysiology of disease and possible mechanisms of retinal involvement [11,12,13].

This review aims to summarize and discuss the various hypothesized pathophysiological mechanisms leading to retinal vessels involvement in *COVID-19* patients. A brief description of clinical retinal findings due to *SARS-CoV-2* present in the literature thus far will also be discussed in order to better understand the various proposed pathogenetic mechanisms.

## 2. Methods

A systematic literature search in the PubMed and Google Scholar databases was performed between 15 March 2020 and 10 July 2022 according to the PRISMA guidelines 2020 [14] and the flow diagram shows the search strategy (Figure 1).

The search terms were “retina”, “artery”, “vein”, “pathogenesis”, “vascular”, “*COVID-19*”, “coronavirus”, “*SARS-CoV-2*” in various combinations along with Boolean search operators “OR” and “AND”. References were also checked for further articles meeting the search criteria. Articles not written in English were excluded. Two independent reviewers (EDA and CME) screened titles and abstracts for articles. To be included in this review, the articles had to meet the following inclusion criteria: prospective or retrospective studies, reviews, case series or case reports reporting retinal vascular manifestations in *COVID-19* patients independent of age, gender, and race. Subjects needed to present clinical symptoms or a positive RT-PCR test for *COVID-19*. Also included were publications reporting histopathologic findings on post mortem donor eyes as well as fundamental in vitro or in vivo research on the pathogenetic mechanism of *COVID-19* disease. Full text of articles that met the inclusion criteria were downloaded and data extracted by the two reviewers independently. In case of discrepancy, a third reviewer was consulted (AK).

Clinical findings were divided into three groups according to the retinal manifestations. In particular, microvascular changes, occlusion of the retinal artery or vein as well as pathogenetic mechanisms explaining these clinical manifestations were the study outcomes.

## 3. Retinal Findings in *COVID-19* Patients

Through the electronic databases, we identified 546 citations. After removing 389 duplicate records, screening of titles and abstracts was done for 157 papers. We performed full text screening for 59 articles. Twenty-one papers met the inclusion criteria and were used for data extraction. Among the final included manuscripts, there was one case control study, 7 case series, 9 case reports, and 4 post-mortem pathology studies. The PRISMA flow chart shows the full screening process (Figure 1).

Demographic and clinical manifestations of retinal findings are summarized in Table 1, Table 2 and Table 3.

### 3.1. Retinal Microvasculopathy

The first paper to report retinal findings in patients with *COVID-19* was published in 2020 by Marinho et al. [15]. Eleven patients diagnosed with *COVID-19* were examined by fundus examination, fundus photography and optical coherence tomography (OCT) at 11 to 33 days after *COVID-19* symptom onset, that did not require intensive care. On fundus examination, four of the eleven patients presented cotton wool spots and micro hemorrhage along the retinal arcades. These retinal vascular abnormalities, while indicative of inner retinal vasculopathy, are characteristically observed in patients with hypertension and diabetes. A main criticism of this study has been that comorbidities of the 11 patients were not mentioned, so it is difficult to link the retinal findings with *COVID-19* [16].

The SERPICO-19 study [17] was a cross-sectional, monocentric, exploratory study for retinal findings, which included consecutive patients diagnosed with *SARS-CoV-2* infection via a nasopharyngeal swab and admitted to the infectious disease department who had symptom onset within 30 days from fundus examination. Subjects with a previous intensive care unit stay and those with known retinal disorders were excluded. Fifty-four *COVID-19* patients were included. All patients underwent dilated fundus examination and photography. Retinal hemorrhages (9.25% vs. 1.5%, *p* = 0.01) and cotton wool spots (7.4% vs. 0%, *p* = 0.006) were significantly higher when compared to age matched unexposed subjects. Furthermore, dilated veins were observed in 15 patients (27.7% vs. 3.0%, *p* = 0.0001). In the follow-up study [18] conducted 6 months later, these abnormalities were significantly reduced compared to baseline (retinal hemorrhages 1.7%, *p* = 0.0009, cotton wool spots 0%, *p* = 0.5, dilated veins 6.8%, *p* = 0.007). The authors proposed that these retinal signs were due to an “impairment to the retinal microcirculation induced by the transient hypercoagulability and the endothelial dysfunction typical of the acute phase of the disease”.

Invernizzi et al. [17,18] measured retinal vessels diameters through a semi-automatic analysis of fundus photography and found that mean diameters of arteries and veins located between 0.5 and 1 disk diameter from the optic disc margin were significantly higher in *COVID-19* patients compared to normal subjects. Furthermore, the mean venular diameter was directly correlated with disease severity, and negatively correlated with the interval between systemic onset of symptoms and the day of fundus photography. At follow-up examination, the vessel diameter (both arterial and venular) was significantly decreased from baseline (*p* < 0.0001). This was not observed in unexposed subjects. Moreover, those who suffered from severe *COVID-19* still presented thicker arteries when compared to non-severe *COVID-19* and unexposed participants, suggesting that patients who suffered from severe *COVID-19* might have long lasting retinal vessel dilation even after resolution of the disease. The authors hypothesized this to be due to a structural damage to the vessel wall either from direct action of the virus or from the cytokine storm in response to viral infection. Aydemir et al. [19] also reported similar increases in mean arterial and venular diameters in *COVID-19* patients.
biomedicines-10-02710-t001_Table 1Table 1Demographic and clinical manifestations of patients with retinal microvasculopathy.Author, Month, YearStudy DesignLocation CountryAge, GenderTotal EyesDiagnosisCOVID StatusBiochemistryComorbidityMarinho PM et al [14],May, 2020Case seriesBrazilRange 25–7011Cotton wool spot and retinal microhemorrhages (n 4)RT-PCR+ (n 9)Antibody+ (n 2)NANAInvernizzi A, et al [16],Oct, 2020Case seriesItaly50 (range 23–82)54Cotton wool spot (7.4%)Retinal hemorrhages (9.25%)Dilated veins(11.1%)Tortuous veins (12.9%)RT-PCR+PT (ratio) 1.19, PTT(ratio) 1.16Fibrinogen (mg/dL)550.2CRP (mg/L) 26.2Ferritin (mg/L) 662LDH (U/L) 270.9Hypertension, diabetes, dyslipidemia,strokeAydemir E, et al [18],July, 2021Case control seriesTurkeyMean 3846 casesDilated arteries and veinsRT-PCR+NANAPereira LA, et al [19],May 2020Case seriesBrazilMedian 62.518Cotton wool spot (16.7%)Retinal flame-shaped hemorrhages(22.2%)RT-PCR+ICU (94.4%)NAHypertension, diabetesRT-PCR: real time polymerase chain reaction; IUC: intensive unit care; NA: not applicable; CRP: C-reactive protein; LDH: lactate dehydrogenase.

Pereira et al. [20] described the presence of flame-shaped hemorrhages (in 22.2% of patients) and cotton wool spots (16.7%) as being the main findings in their cohort of 18 patients who were hospitalized for severe *COVID-19*. However, in this study, the authors clearly state that patients with hypertension and diabetes were not excluded, as their study would have thus been unfeasible, making the characterization of a typical *SARS-CoV-2* retinopathy difficult.
biomedicines-10-02710-t002_Table 2Table 2Demographic and clinical manifestations of patients with retinal vein occlusion.Author, Month, YearStudy DesignLocation CountryAge, GenderTotal EyesDiagnosisCOVID StatusBiochemistryComorbiditySheth JU et al [20],July, 2020Case reportIndia52, M1HRVORT-PCR +No hospitalizationNAnoFonollosa A et al [21],March, 2022Case seriesSpain39 (range 30–67)18CRVO (n 9)BRVO (n 4)CRVO + CRAO (n 3)CRAO (n2)RT-PCR+ (n 18)Asymptomatic (n 3)Mild disease no hospitalization (n 8)ICU (n 4)Elevated D-dimer and plateletsHypertension, diabetes, dyslipidemiaInvernizzi A et al [22],June, 2020Case reportItaly54, F1Impending CRVORT-PCR+PT 13.8 s (INR 1.27)aPTT 36.6 s (RATIO1.26)Fibrinogen 6.82 g/lD-dimer 426 lg/LNAInsausti-Garcia A et al [23],June, 2020Case reportSpain40, M1CRVO with papillophlebitisIgM+ IgG+ *SARS-CoV-2* (ELISA)D-dimer 672 µg/LFibrinogen 451 mg/dLNAGaba WH et al [24],June, 2020Case reportUAE52, M1Bilateral CRVO + optic disc edemaRT-PCR+Severe pneumoniaDVT right leg (Doppler US)Severe dilation of right ventricle (echocardiography)D-dimer >20 µcg/mLLDH 402 IU/LFerritin 1518 µg/LNAShroff D, et al [25],Mar, 2022Case seriesIndiaRange 32–814CRVO, BRVO, CRAO, vitreous hemorrhagePT-PCR+HospitalizationICU (n 3)Elevated D-dimer and fibrinogennoShiroma HF, et al [26],June, 2022Case seriesBrazil48 (range 27–73)14CRVO (n 8)HRVO (n 1)BRVO (n 3)CRAO (n 2)PT-PCR+Hospitalization (n 3)Elevated D-dimer and C-reactive proteinHypertension, diabetes, dyslipidemiaAshkenazy N et al [27],March, 2022Case seriesUSA32(range 18–50)12CRVO (n 9)HRVO (n 3)PT-PCR+Hospitalization (n 3)NAnoHRVO: hemi-retina vein occlusion; CRVO: central retina vein occlusion; BRVO: branch retina vein occlusion; CRAO: central retina artery occlusion; RT-PCR: real time polymerase chain reaction; IUC: intensive unit care; DVT: deep venous thrombosis; NA: not applicable; CRP: C-reactive protein; LDH: lactate dehydrogenase.

### 3.2. Retinal Venous Occlusion

Our search of the literature regarding *COVID-19* retinal venular or arterial occlusions revealed that central retinal vein occlusion (CRVO) was the most often described pathology. Of note, a case of inferior hemiretinal vein occlusion in an otherwise healthy patient was described by Sheth et al. [21]. OCT and fluorescein angiography showed the presence of macular edema and vessel wall staining, respectively, suggesting a vasculitic occlusion. Fonollosa et al. [22] reported a series of 18 eyes with different types of occlusions (9 CRVO, 4 branch retinal vein occlusion (BRVO), 3 combined arterial and vein, 2 central retinal artery occlusion (CRAO)) in 15 patients with different grade of severity of *COVID-19* disease (3 asymptomatic, 8 with mild disease, 4 hospitalized for severe respiratory disease). Interestingly, usually the non-COVID subjects with retinal vein occlusion are older than 60 years with cardiovascular disease, while the median age of this cohort was 39 years and only 36% of these *COVID-19* patients presented cardiovascular disease [22]. The clinical manifestations of patients with retinal vein thrombosis are variable. Invernizzi et al. [23] reported an impending CRVO without macular edema in a 54-year-old female with elevated D-dimer (426 µg/L) and fibrinogen (6.82 g/L). Another patient presented as CRVO with papillophlebitis 6 weeks after *SARS-CoV-2* infection [24]. The only reported bilateral CRVO with optic disc edema was a 40-year-old man hospitalized for severe *COVID-19* pneumonia and associated deep vein thrombosis and severe dilation of the right ventricle on echocardiography [25]. Shroff et al. [26] reported four cases of retinal vascular disease (1 CRVO, 1 BRVO, 1 CRAO, and 1 vitreous hemorrhage) developed one month post hospitalization for moderate to severe *COVID-19* infection in otherwise healthy subjects. All patients presented high levels of D-dimer and fibrinogen. Similarly, in a case series of 14 patients from multiple centers in Brazil [27], the mean age was 48 years and 86% had no risk factors for cardiovascular disease, but they were all tested positive for *COVID-19* infection and three of them hospitalized for severe disease. Eight presented CRVO, 1 hemiretinal RVO, 3 BRVO, and 2 CRAO. Recently, Ashkenazy et al. [28] reported a case series of 12 patients that developed CRVO (9 of 12 eyes) or HRVO (3 of 12 eyes) after *COVID-19* infection. Three patients were hospitalized but not intubated. Exclusion criteria were the risk factors most often associated to retinal vein occlusion in the general population and in particular, age older than 50 years, any risk factor for cardiovascular disease, diabetes, thrombophilia, coagulopathies, and glaucoma. The occurrence of thromboembolic events in this selected young otherwise healthy *COVID-19* population shows the important role of inflammatory cytokines and activation of coagulative state in the pathogenesis of this condition.

### 3.3. Retinal Arterial Occlusion

The cases of retinal artery occlusion reported in the literature are rare and of the non-arteritic form. Ucar et al. [29] presented a case of central retinal artery occlusion (CRAO) in a patient with no known cardiovascular risk factors. Kulkarni et al. [30] reported one case of choroidal artery occlusion and one case of both central retinal artery and vein occlusion in patients hospitalized for *COVID-19* infection. Similarly, Acharya et al. [31] described the case of a 60-year-old man hospitalized in the intensive care unit for severe respiratory disease who experienced CRAO 12 days later. The levels of D-dimer, fibrinogen and IL-6 were elevated. Three patients presented with paracentral scotoma associated with inner and outer retina layers focal disruptions suggestive of paracentral acute middle maculopathy (PAMM) and acute macular neuroretinopathy (AMN). These findings are usually an expression of dysfunction of arterial microvessels. [32,33] However, it remains difficult to identify the *SARS-CoV-2* as the sole cause of the occlusion, due to the presence of confounding factors in the majority of the case reports. Montesel et al. [34] reported a CRAO in a 59-year-old African man 10 days after hospitalization in ICU for severe respiratory distress due to *COVID-19* infection. D-dimer and fibrinogen levels were elevated. As the patient had a history of hypertension, it was not possible to exclude a multifactorial etiology. Based on the current understanding of the pathophysiology of *COVID-19*, altered inflammatory and coagulation parameters are crucial in assessing the causality at the time of the vascular event, while comorbidities and age of the patients might be additional risk factors for retinal vascular complications and thus confound the association between the viral infection and the retinal event.
biomedicines-10-02710-t003_Table 3Table 3Demographic and clinical manifestations of patients with retinal artery occlusion.Author, Month, YearStudy DesignLocation CountryAge, GenderTotal PatientsDiagnosisCOVID StatusBiochemistryComorbidityKulkarni MS et al [29], May, 2022Case reportIndia49, M20, M2Choroidal artery occlusionMixed CRAO + CRVO RT-PCR+HospitalizationRT-PCR+Hospitalization NAD-dimer: 321 ng/mL, C-reactive protein (CRP; 2.44 mg/dL), lactate dehydrogenase (LDH; 414 IU/L)NANAAcharya S et al [30], June, 2020Case reportUSA60, M1CRAORT-PCR+ICUD-dimer: 42.131Fibrinogen: >700C-reactive protein: 7.02Ferritin: 324Procalcitonin: 0.07IL-6: 546.1NAGascon P et al [31], May, 2020Case reportFrance53, M1PAMM, AMN, deep retinal hemorrahagesRT-PCR+Chest CT: bilateral ground glass opacitiesC-reactive protein: 29 mg/LthrombocytosisNAVirgo P et al [32],June, 2020Case reportUK37322PAMM, AMNRT-PCR+IgG+NANANANAMontesel A et al [33], Dec, 2020Case reportSwitzerland59, M1CRAORT-PCR+ICUD-Dimer: 2059 ng/mL (normal <500 ng/mL);Fibrinogen: 5.9 g/L (normal 2.0–4.0 g/L);C-reactive protein: 184 mg/L (normal <10 mg/L);IL-6: 42 pg/dl (normal <11 pg/dL)Hypertension, hyperuricemiaCRVO: central retina vein occlusion; CRAO: central retina artery occlusion; PAMM: paracentral acute middle maculopathy; AMN: acute macular neuroretinopathy; RT-PCR: real time polymerase chain reaction; IUC: intensive unit care; DVT: deep venous thrombosis; NA: not applicable; CRP: C-reactive protein; LDH: lactate dehydrogenase; IL-6: interleukin-6.

## 4. Detection of *SARS-CoV-2* and its Receptors in Human Retina

Casagrande et al. [11] were the first to report the presence of *SARS-CoV-2* nucleic acid in the human retina. Specimens were tested via RT-PCR. 21% of the retinal biopsies of deceased *COVID-19* patients were found to be positive for the virus. Subsequently, Araujo-Silva et al. [35] observed presumed spike protein and nucleocapsid *COVID-19* proteins by immunofluorescence microscopy in “endothelial cells close to capillary flame and cells of the inner and outer nuclear layers. By transmission electron microscopy double-membrane vacuoles that are consistent with the virus were seen in the perinuclear region of these cells”. Through these histological studies, it is clear that *SARS-CoV-2* can reach retinal tissue and therefore cause damage. How the virus enters the retinal cells is not well understood, but numerous theories exist. Saggiora de Figueiredo et al. [36] have hypothesized infection *per continuitatem* from the anterior segment, hematogenous and neuronal routes.

For *SARS-CoV-2* to access host cells, its surface glycoprotein, known as spike, binds to the protein ACE2 receptor [37]. The ACE2 receptor is part of the renin-angiotensin system, which is crucial in regulating blood pressure, fluid, and electrolyte homeostasis. Other receptors have also been suggested to mediate entry of *SARS-CoV-2* such as transmembrane serine protease 2 (TMPRSS2) [38] and sialic acid receptor [39]. These proteases play an important role in the entry of the virus into the target cells by interacting with the spike protein. In particular, in the *SARS-CoV-2* virus, a single region of the spike protein, called the receptor-binding domain (RBD), mediates interaction with the host cell receptor. Following receptor binding, a target cell protease cleaves the spike that releases the spike fusion peptide, thereby facilitating virus entry [39] (Figure 2).

Zhou et al. [40] investigated the expression of ACE2 receptor and TMPRSS2 in human retina. ACE2 receptor was found in both non-vascular neuroretinal cells in the retinal ganglion cell layer and inner plexiform layer, cells of the inner nuclear layer (horizontal cells, bipolar cells, amacrine cells, interplexiform neurons, Müller cells), photoreceptor outer segments, and in capillary endothelial cells. TMPRSS2 was also identified in some cells of the neuro-retina, in Mueller glia, and in vascular/perivascular cells. Both receptors have also been identified by Schnichels et al. [41]. A recent immunohistopathologic study [12] on 11 eyes from *COVID-19* donors showed the presence of *SARS-CoV-2* S protein within the retina close to the optic nerve, while this was not evident in control donor eyes. The identification of both the *SARS-CoV-2* virus and its host receptors in the human retina supports the notion of a susceptibility of the retina to *COVID-19* disease. However, the amount of virus and receptor expression required to cause clinical manifestation is still not known.

## 5. Pathophysiological Mechanisms

Recent evidence strongly indicates *COVID-19* to be a vascular disease by affecting both arteries and veins. However, the pathogenetic mechanism is not clear yet. A prothrombotic state may be triggered by various mechanisms, such as endothelial and vascular injury or inflammation.

The endothelium might be affected via the ACE2 receptor [42]. The endothelium naturally prevents blood clotting, but when invaded by *SARS-CoV-2*, it loses its ACE2 activity, leading to an increase in angiotensin II which in turn promotes vasoconstriction and thrombogenicity by increasing platelet and leucocyte adhesion [43]. Recently, Reinhold and associates [44] identified endothelial damage and microthrombi in eight out of ten eyes of patients who had died from *COVID-19*, corroborating this mechanism in the eye.

Characteristic of *COVID-19* associated coagulopathy is increased prothrombotic markers, such as fibrinogen and D-dimer levels [45]. More specifically, high D-dimer levels have been associated with both venous and arterial thrombosis even in absence of previous history of vascular disease and are a factor for poor prognosis [46]. D-dimer was reported to be elevated in 61% of ocular vascular occlusions with 62% of patients diagnosed with an asymptomatic or mild *COVID-19* disease [22]. This has been described in a case of CRAO in an otherwise healthy male described by Ucar et al. [29], with D-dimer level being reported to be 1041 ng/mL (normal range: 80–630 ng/mL) and fibrinogen was 405.1 g/L (normal range: 180–400 g/L). Similarly, Montesel et al. [34] described a case of CRAO due to *SARS-CoV-2* in a patient known for hypertension and hyperuricemia. In this case, the D-dimer was 2059 ng/mL (normal: <500 ng/mL) and fibrinogen levels were 5.9 g/L (normal: 2.0–4.0 g/L). Dumitrascu and colleagues also described a case of incomplete ophthalmic artery occlusion (OAO) in a patient with severe *COVID-19* who was on therapeutic anticoagulation with apixaban for deep venous thrombosis that occurred during his hospitalization in intensive care unit for a severe form of *COVID-19* infection. D-dimer levels at presentation were 2.13 microg/mL (normal ≤ 0.05 mg/mL FEU) and fibrinogen was 608 mg/dL (normal: 200–400 mg/dL) [47].

Further evidence of the role of prothrombotic markers can be found in a case of impending CRVO in an otherwise healthy *COVID-19* patient [23]. A 54-year-old woman presented with impending CRVO and at the same time was diagnosed with mild *COVID-19* with pneumological involvement. Her serologic examination revealed elevated fibrinogen levels at 6.82 g/L (normal: 1.70–4.00 g/L), D-dimer was 426 microg/L FEU and PT (13.8 s, INR 1.27, normal range 0.85–1.18) and aPTT (36.6 s, RATIO 1.26, normal range 0.85) were also elevated. Due to her respiratory symptoms, she was treated with methylprednisolone 1 g IV Three days later, the coagulation indexes were normalized. The authors speculated that, by having treated the patient with prednisone, the coagulation and inflammatory status might have been normalized with restoration of normal blood flow.

A dysregulated immune system, such as that induced by *SARS-CoV-2*, also plays a role in endothelial dysfunction and thrombosis. Reports of hospitalized *COVID-19* patients have identified what is known as a cytokine storm: an excessive production of proinflammatory cytokines such as TNF-alpha and IL-6 [48]. This hyper-inflammation further causes endothelium cell damage, and increases proinflammatory gene expression, recruiting inflammatory cells and altering the thrombotic homeostasis, leading to impairment of the microvasculature [49]. Inflammatory cells such as mononuclear and neutrophilic cells were found in the lumen and intima of congested capillaries of the choriocapillaris in both eyes of four patients deceased from COVID [44]. Jidigamet al. reported a loss of retinal microvasculature and thinning of the microcapillaries in *COVID-19* donor eyes when compared to the control group [12]. In these eyes, immunohistochemical tests showed an increase in the number of activated microglial cells and astrocytes, which can be caused by direct or systemic inflammatory process. However, no evidence of *SARS-CoV-2* associated inflammation in the neuroretina was identified. More histological studies are needed to further elucidate this process.

In the SERPICO follow-up study [18] Invernizzi et al. found that most of retinal vasculature alterations identified in patients with *COVID-19* regressed, with mean artery and vein diameter reducing significantly (*p* < 0.0001). They hypothesized that the dilation during disease could be either due to direct damage to the vessel wall by the virus or due to transient modification induced by the cytokine storm. Endothelial cells in both retinal arteries and veins can express inflammatory cytokine receptors and dilate in response to the stimulus [50]. In the SERPICO-19 follow-up study [18], all *COVID-19* patients had normalized inflammatory and coagulation indexes supporting the hypothesis that retinal vessel diameter is related to inflammatory status of the patient. Patients with more severe symptoms initially, however, remained partially dilated compared to those of non-severe and unexposed subjects. This could be correlated to a larger endothelial activation that is correlated with disease severity.

## 6. Discussion

Since the beginning of the pandemic outbreak in December 2019, more than 6.5 million of people have died (www.covid19.who.int (accessed on 19 September 2022)) from complications of *COVID-19* disease. Although the *SARS-CoV-2* affects primarily the lungs, other organs are often involved, such as skin, central nervous system, and gastrointestinal tract, as well as the eyes. Ocular manifestations vary from anterior segment involvement such as conjunctivitis and intraocular inflammation to vision threatening retinal and optic nerve involvement, such as optic neuritis and retinal vascular occlusion. In this review, we have summarized the case reports and case series describing the various retinal thromboembolic manifestations observed in patients with *COVID-19* and tried to understand their pathogenetic mechanisms based on clinical and histologic evidence.

We have categorized the various retinal vascular findings in three major groups, and specifically retinal microvasculopathy, retinal venous occlusion, and retinal arterial occlusion. Four case series reported mild changes at the level of the retinal microvasculature like cotton wool spots, focal and flam-shaped hemorrhages, and vein dilation in 129 eyes. All patients have been tested positive for *COVID-19* (RT-PCR), while only in one series biochemistry results were reported showing high levels of coagulation factors, fibrinogen, and CPR. Comorbidities, such as diabetes and hypertension, were not always reported, representing a limitation in the interpretation of the findings described (Table 1).

The second group of manifestations concerns occlusion of the retinal vein. Four case reports and four case series described branch, hemiretinal, or central retina vein occlusion in a total of 52 eyes. All subjects presented a positive test for *COVID-19* and the majority were hospitalized (7 in ICU). Elevated D-dimer, platelet, and fibrinogen were reported except that for 13 cases, while there is a general lack of information about comorbidities (Table 2).

For the third category of occlusion arterial changes, we assigned five case reports ranging from PAMM and AMN in 3 eyes to central retinal artery occlusion in 4 eyes. All patients were *COVID-19* positive according to RT-PCT test, and four were hospitalized (2 in ICU). D-dimer, fibrinogen, CRP were elevated in all subjects except for two (not reported), while in two cases IL-6 was tested and resulted outside normal range. Unfortunately, also for these patients the comorbidities were not reported except for one subject that presented hypertension (Table 3).

The revision of these articles shows that a prothrombotic state seems to be a present in all affected subjects, independently from the clinical manifestation.

The pathogenesis of the many ways that *COVID-19* causes organ damage is still unclear, although it seems to be multifactorial. Two possible mechanisms have been proposed to explain vascular damage in *COVID-19* disease affecting either arterial or venous circulation. First, a hypercoagulable state might result from direct viral-mediated endothelial damage [28]. In fact, the virus invades into the cell through the ACE2 receptor and activates the renin-angiotensin system (RAS) [42,43]. This activation determines a loss of ACE2 and a consequent increase in Angiotensin II, which leads to vasoconstriction and increases platelet and leucocytes inducing thrombogenicity. The endothelial cell damage upregulates proinflammatory cytokines. This hypothesis is supported by clinical evidence of increased levels of D-dimer and fibrinogen in almost all patients with retinal vascular occlusion reported in the literature. Moreover, post-mortem analysis described the expression of ACE2 receptor in several tissues, including the retinal endothelial cells [11].

This mechanism may explain, in part, why retinal vascular occlusions might affect otherwise healthy subjects and be limited to the retinal vasculature without other thrombotic events in other territories possibly due to viral tropism for retinal vasculature [22].

The second pathogenetic mechanism, the cytokine cascade activation, is responsible for a disseminated intravascular coagulation-like state, also known as “cytokine storm”, causing endothelial damage and vascular occlusion. Clinical and experimental evidence support this hypothesis. In particular, an animal model of experimental coronavirus retinopathy (ECOR) has demonstrated that the virus infiltration causes retinal vasculitis, breakdown of the blood-retinal barrier, and retinal degeneration [51] with extravasation of erythrocytes and immune cells into the surrounding tissue. This interstitial inflammation might result into a vasculitis, leading to hyaline thrombi. Although retinal findings in post-mortem studies have been inconsistent, fibrin microthrombi and inflammatory cells were observed at the level of retinal and choroidal vessels of eyes from patients who died from *COVID-19* [15].

The ECOR model gives additional information concerning the biphasic behavior of Coronavirus, with a first primary infection triggering the immune system with activation of immune cells and release of inflammatory mediators, which then play a crucial role in the second phase that looks like an autoimmune disease. This biphasic manner is commonly observed in the progression of the severity of the clinical status [52].

To understand the pathogenesis and the mediators involved is fundamental for the treatment and the control of the disease. Moreover, retinal findings might indicate a predominantly hypercoagulable state, which requires anticoagulant drugs, rather than an inflammatory response requiring steroids and inhibitors of interleukines such as IL-6 according to the phase of the disease. For this reason, it would be useful to always perform a fundus examination of patients with *COVID-19* in order to detect early asymptomatic retinal signs that could guide the treatment. The non-arteritic retinal artery occlusions are usually secondary to an atherosclerotic lesion or hypercoagulable state. Since the retina is not the only human tissue to present with arterial thrombotic events, with cases of acute coronary occlusions, stroke, and limb ischemia having been reported in *COVID-19* positive patients [53], a prompt anti-coagulant treatment might reduce the development of more severe forms of the disease, speculating on the predominant role of the hypercoagulable condition. However, the exact mechanism of arterial thrombosis is still unclear and there is no evidence of two different pathways involved for vein and artery occlusion.

On the other side, asymptomatic retinal microangiopathy is commonly observed during other viral infections like the human immunodeficiency virus disease or systemic diseases like diabetes and hypertension. Clinical manifestations like cotton wool spots and microhemorrhages might be similar to what observed in *COVID-19* positive subjects. Since inflammation is known to be associated with retinal lesions in patients with diabetes [54], it is reasonable to presume that *COVID-19* may also precipitate these lesions. For this reason, it would be important to be aware of comorbidities as possible confounAding factor. Unfortunately, the reports present in the literature often lack information concerning comorbidities, which renders impossible to establish a direct correlation of *SARS-CoV-2* infection and retinal occlusive findings in the majorities of cases. Moreover, the case-studies analyzed did not include a recent pre-covid fundus examination of the patients, making thus impossible to establish a specific effect of virus infection rather than aging as the dominant etiologic factor.

Finally, despite the several cases published, the incidence of both arterial and venous occlusions secondary to *COVID-19* disease as well as the true prevalence of *COVID-19* among subjects with retinal thromboembolic events is unknown. Large-scale observational studies are needed to answer this question.

**Limitations** of this review are primarily related to the included studies. In fact, they are mostly case reports and observational case series. Larger cohort with longer follow-up and different stage of the disease would be necessary to understand the impact and the natural history of *COVID-19* on retinal vascular occlusions. Several questions remain open, in particular the incidence and prevalence of *COVID-19* on retinal manifestations, the correlation with the severity of the disease, and the role of comorbidities such as diabetes and hypertension. Moreover, the pathogenetic mechanisms remain not well understood and few hypotheses are proposed, supported by clinical observation and post-mortem studies. A specific mechanism different for retinal arterial and venous occlusion has not been described yet. On the other side, we strongly believe that it is important for the ophthalmic community in particular and the care giver community in general, to know and recognize the clinical signs of retinal vascular involvement secondary to *COVID-19* disease and this study will contribute to this knowledge. Moreover, understanding the etiology of the disease would guide the correct treatment with the aim of reducing the severity of the disease.

## 7. Conclusions

In summary, this is an updated systematic review summarizing retinal vascular findings secondary to *COVID-19* disease since the beginning of the pandemic. The retinal manifestations can vary from mild signs like cotton wool spots, microhemorrhages, flame-shaped hemorrhages, and dilated veins, to more severe forms like focal retinal infarcts, branch or central vein or artery occlusion inducing irreversible visual impairment and decrease in quality of life [55]. These manifestations occurred mostly in patients hospitalized for a severe form of *COVID-19* disease. Clinical and histopathologic studies have proven that the retina, and more specifically its vasculature, is affected by *SARS-CoV-2*. Several studies proposed different hypothesis on the mechanisms responsible for these alterations. Direct viral endothelial injury, endothelitis, activation of the immune response by a cytokine storm leading to a procoagulant state are the most likely pathways responsible for the vascular manifestations of the *SARS-CoV-2* disease at the level of the retina. Further studies are needed to elucidate the virus’s pathophysiological mechanism in the retina and confirm the association with the clinical features. More specifically, it is important to identify and characterize the mechanisms of a specific *COVID-19* retinopathy. With this knowledge, a simple fundus examination may aid in the diagnosis, treatment, and follow-up of these patients.

## Figures and Tables

**Figure 1 biomedicines-10-02710-f001:**
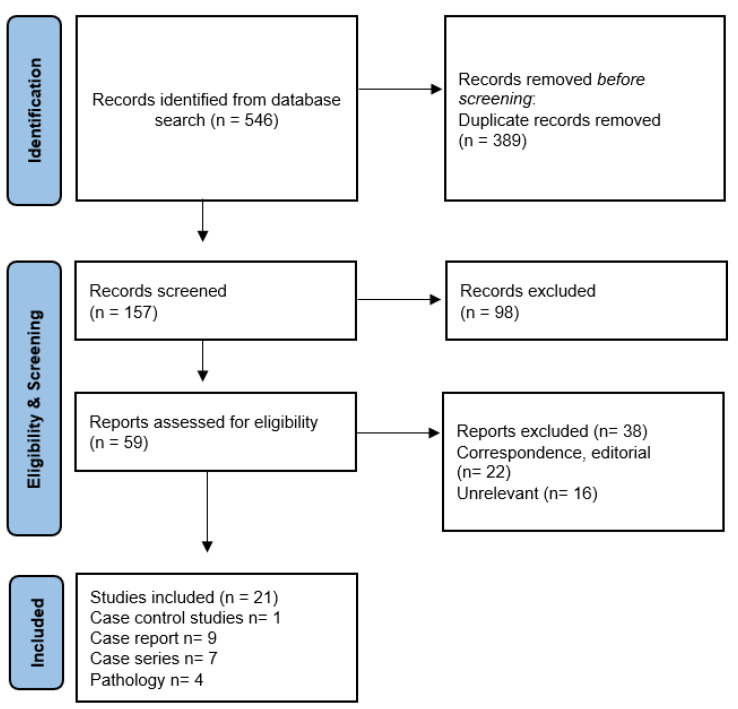
PRISMA 2020 flow diagram showing the search strategy in this systematic review.

**Figure 2 biomedicines-10-02710-f002:**
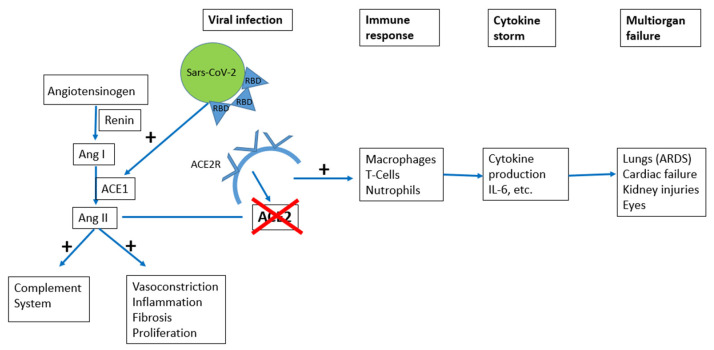
*SARS-CoV-2* access and subsequent immune response. Ang I: angiotensin I; Ang II: angiotensin II; ACE1: angiotensin converting enzyme 1; ACE2: angiotensin converting enzyme 2; ACER2: angiotensin converting enzyme receptor 2; RBD: receptor-binding domain.

## Data Availability

Publicly available datasets were analyzed in this study. This data can be found on Pubmed and Google Scholar search database.

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
