# Peer review of "Pathogenesis of Vascular Retinal Manifestations in COVID-19 Patients: A Review"

_biomedicines, 2022, doi:10.3390/biomedicines10112710_

Round 1

Reviewer 1 Report

The paper entitled „Pathogenesis of vascular retinal manifestations in COVID-19 patients: a review” is focused on artery and (or) vein retina manifestations and the possible pathogenetic mechanisms. I found this paper interesting reading, however, there are a few issues which the authors should consider during revisiting the current version of the manuscript.

- please provide short summary of the results in the abstract;

- have you followed any guidelines in performing systematic literature review? if yes, please provide relevant references;

- please format Table 1 as well as the other Tables, accordingly to the journal guidelines;

- please provide the Discussion Section, including: both theoretical and practical implications of your study, along with the study contributions and limitations (threats to validity);

- please add the Conclusions Section, including the summary of the results, as well as the future research agenda.

To sum up. The current version of the manuscript needs a major revision.

Author Response

The paper entitled „Pathogenesis of vascular retinal manifestations in COVID-19 patients: a review” is focused on artery and (or) vein retina manifestations and the possible pathogenetic mechanisms. I found this paper interesting reading, however, there are a few issues which the authors should consider during revisiting the current version of the manuscript.

- please provide short summary of the results in the abstract;

Answer: the abstract has been modified and a short summary of the results was added. (page 1)

- have you followed any guidelines in performing systematic literature review? if yes, please provide relevant references;

Answer: to perform the systematic literature review we followed the PRISMA 2020 guidelines. We updated the methods section (page 2) with the following sentence: “according to the PRISMA guidelines 2020 [14] and the flow diagram shows the search strategy (Figure 1).” The reference [14] was added in the reference section (page 12).

Figure 1 “PRISMA 2020 flow diagram showing the search strategy in this systematic review” was also added (page 2).

- please format Table 1 as well as the other Tables, accordingly to the journal guidelines;

Answer: Tables have been formatted accordingly to the journal guidelines

- please provide the Discussion Section, including: both theoretical and practical implications of your study, along with the study contributions and limitations (threats to validity);

Answer: A discussion section according to your suggestion was added (page 9 - 11). The “limitations” paragraph was rephrased (page 11).

- please add the Conclusions Section, including the summary of the results, as well as the future research agenda.

Answer: The Conclusions section was added (page 11 -12).

Reviewer 2 Report

1.      What are the main findings for each table summarized  in section 3?

2.      Is there any difference in the COVID pathophysiological mechanisms between the retina itself and its vasculature? How does the retinal vasodilation activity change if the patient is infected with COVID?

Grammar syntax needs to be carefully checked.

Author Response

  1. What are the main findings for each table summarized in section 3?

 Answer: the main findings for each table summarized in section 3 are reported in the discussion section (page 9 - 10).

  1. Is there any difference in the COVID pathophysiological mechanisms between the retina itself and its vasculature? How does the retinal vasodilation activity change if the patient is infected with COVID?

 Answer: Following the suggestion of Reviewer 1, we added a Discussion section (page 9 – 11). The pathophysiological hypothesis and the clinical and histological evidences supporting these hypotheses are reported in the “Pathophysiological Mechanisms” section (page 8 – 9) and discussed in the Discussion section (page 10, from 3rd paragraph). Since we focused our study on retinal vascular findings, we didn’t search and reported other possible mechanisms related to other retinal or ocular manifestations.

The retinal vasodilation activity changes after COVID-19 infection is described among others, in the SERPICO study that we reported on page 4, 2nd and 3rd paragraph, and page 9, 2nd paragraph. Moreover, a possible mechanism explaining the vascular changes is represented by the activation of the renin-angiotensin system (RAS), described in the discussion section on page 10, 3rd paragraph) and figure 2 (page 7).

Grammar syntax needs to be carefully checked.

 Answer: An extensive grammar revision was performed by English native professionals.

Reviewer 3 Report

This review covers the COVID-19-associated ocular manifestations affecting the posterior segment of the eye. Given the paper reviews a topic from recent years, there is little in the literature (mostly case report studies) to form a solid consensus about some of the secondary effects of infection but remains an important area of study. Overall, this is a well-written review article that provides a succinct overview of this newer and unexplored topic. Suggestions are made to improve the quality of the manuscript.

-The authors describe this article as a systematic review. The PRISMA guidelines should be reviewed to verify that the paper is inclusive of the generally required elements in order to be considered a 'systematic review.' In particular, where these outcomes pre-selected (10a in the PRISMA 2020 checklist)?

-There is a focus on (3) different clinical manifestations that have been reported. Are there other ocular manifestations than these? Please clarify in the manuscript.

-A mechanism figure describing the interactions between ACE2 and inflammatory pathways would be beneficial to the readership.

-Please modify the 'Proposed Patho. Mechanism' section to focus solely on mechanism with removal of the named case report studies.

-Mention of whether these case-studies included a recent pre-covid fundus evaluation for determining a specific effect of infection, rather than aging. Co-morbidities present should also be included if known. 

-Please include the incidence of these ocular manifestations following covid infection.

Author Response

This review covers the COVID-19-associated ocular manifestations affecting the posterior segment of the eye. Given the paper reviews a topic from recent years, there is little in the literature (mostly case report studies) to form a solid consensus about some of the secondary effects of infection but remains an important area of study. Overall, this is a well-written review article that provides a succinct overview of this newer and unexplored topic. Suggestions are made to improve the quality of the manuscript.

-The authors describe this article as a systematic review. The PRISMA guidelines should be reviewed to verify that the paper is inclusive of the generally required elements in order to be considered a 'systematic review.' In particular, where these outcomes pre-selected (10a in the PRISMA 2020 checklist)?

 Answer: As previously stated, to perform the systematic literature review we followed the PRISMA 2020 guidelines. We updated the methods section (page 2) with the following sentence: “according to the PRISMA guidelines 2020 [14] and the flow diagram shows the search strategy (Figure 1).” The reference [14] was added in the reference section (page 12). The outcomes were pre-selected, according to the Item #10a in the PRISMA 2020 checklist. The inclusion criteria are listed in the Methods section (page 3, first paragraph).

-There is a focus on (3) different clinical manifestations that have been reported. Are there other ocular manifestations than these? Please clarify in the manuscript.

 Answer: We focused our review on retinal vascular findings secondary to COVID-19 disease. However, other ocular manifestations have been reported in the literature as already stated in the Introduction section (page 1, first paragraph) and in the Discussion section (page 9, first paragraph).

-A mechanism figure describing the interactions between ACE2 and inflammatory pathways would be beneficial to the readership.

 Answer: As suggested, a figure describing the interactions between ACE2 and inflammatory response has been added (Figure 2, page 7)

-Please modify the 'Proposed Patho. Mechanism' section to focus solely on mechanism with removal of the named case report studies.

 Answer: Following your comment, we modified the title “Proposed Pathophysiological Mechanisms” to “Pathophysiological Mechanisms”. In this section we reported several studied that support the pathogenetic hypothesis. We also added a Discussion section, which includes, among others, theorical and practical implications of the proposed pathophysiologic mechanisms. We hope that you agree with this new structure that improves the quality of the manuscript  

-Mention of whether these case-studies included a recent pre-covid fundus evaluation for determining a specific effect of infection, rather than aging. Co-morbidities present should also be included if known. 

Answer: Unfortunately, the case-studies did not include any information regarding a recent pre-COVID fundus examination. In the Discussion section (page 11, 2nd paragraph) we added the following sentence: “Moreover, the case-studies analyzed did not include a recent pre-covid fundus examination of the patients, making thus impossible to establish a specific effect of virus infection rather than aging as the dominant etiologic factor”. Also data concerning co-morbidities are very scarce in the case reports. We discussed this point in the Discussion section (page 11, 2nd paragraph). The lack of these information was added in the “limitation” paragraph (page 11).

-Please include the incidence of these ocular manifestations following covid infection.

Answer: The incidence of these specific manifestations is not known to our knowledge. There are a few data concerning the vascular occlusions in other organs, but not only focused on retinal vasculature. We believe that the nature of the reports limits the possibility to have this information. Large-scale studies with longer follow-up, including not principally hospitalized patients, but all COVID-19 positive subjects could respond to this question. In the Discussion section (page 11, third paragraph) we added the following sentence: “Finally, despite the several cases published, the incidence of both arterial and venous occlusions secondary to COVID-19 disease as well as the true prevalence of COVID-19 among subjects with retinal thromboembolic events is unknown. Large-scale observational studies are needed to answer this question.”

Round 2

Reviewer 1 Report

Dear Authors.

Thank you for addressing all the issues raised in my review.

In my opinion, the current version of your manuscript can be considered for the publication.